# Adapting Pretrained Text-to-Text Models for Long Text Sequences

**Wenhan Xiong**[*], **Anchit Gupta**[*], **Shubham Toshniwal,**
**Yashar Mehdad**, **Wen-tau Yih**

Meta AI

{xwhan,anchit,shtoshni,mehdad,scottyih}@fb.com

## Abstract

We present an empirical study of adapting an existing pretrained text-to-text model for long-sequence inputs. Through a comprehensive study along three axes of the pretraining pipeline – model architecture, optimization objective, and pretraining corpus, we propose an effective recipe to build long-context models from existing short-context models. Specifically, we replace the full attention in transformers with *pooling-augmented blockwise attention*, and pretrain the model with a masked-span prediction task with spans of varying lengths. In terms of the pretraining corpus, we find that using randomly concatenated short-documents from a large open-domain corpus results in better performance than using existing long document corpora, which are typically limited in their domain coverage. With these findings, we build a long-context model that achieves competitive performance on long-text QA tasks and establishes the new state of the art on *five* long-text summarization datasets, often outperforming previous methods with larger model sizes.

## 1 Introduction

NLP applications like summarization and question answering often require processing long text sequences. While there have been tremendous empirical breakthroughs (Vaswani et al., 2017; Devlin et al., 2019) from large pretrained language models (PLMs), most of these successes have been confined to *short-context* tasks (Rajpurkar et al., 2016; Wang et al., 2019). On long-context NLP benchmarks (Kočiský et al., 2018; Zhong et al., 2021; Pang et al., 2022b), where the input sequences are often longer than 10,000 tokens, there is still a significant gap between human performance and the state-of-the-art models.

Extending the success of PLMs to long texts is nontrivial for the following reasons. First, the quadratic complexity of self-attention makes it prohibitive to directly apply full-attention to long sequences. Any long-range architecture needs to be computationally efficient and at the same time capture long-distance dependency.[1] Second, the training objectives used by existing PLMs have largely focused on short text and have not been well-studied for long-context scenarios. For instance, BART (Lewis et al., 2020) pretraining involves reconstructing the whole corrupted input sequence, which is impractical for long sequences given the computational overhead of decoder-side attention. Additionally, while abundant short documents can be easily collected from web dumps to pretrain short-context models that work well across different domains, long documents are much scarcer and are often collected from specific domains as books or movie scripts (Gao et al., 2021). It is unknown whether the existing corpora are more effective for pretraining a versatile long-context model compared to using artificially constructed long texts.

In this work, we conduct a thorough experimental study to find a recipe for building high-performing long-context models. In contrast to a recent work (Guo et al., 2022) that pretrains a long-context model from scratch, we choose to adapt an existing short-text model for long texts with further pretraining. Our empirical results demonstrate the effectiveness of this strategy by achieving stronger performance on various downstream tasks, while saving on the high cost of pretraining from scratch. More specifically, we explore three axes of the pretraining pipeline, namely *efficient long-range model architectures*, *long text corpora creation* and

---

[*] Equal Contribution.
[–] Our code has been released at https://github.com/facebookresearch/bart_ls.

---

[1]While there exists a long list of efficient attention variants (Tay et al., 2020), their efficacy is only validated in synthetic or small-scale experiments and it is unknown whether these variants are scalable and suitable for large-scale pretraining for natural language (Xiong et al., 2022; Tay et al., 2022).

*the choice of pretraining objectives*. Our main findings are summarized as follows:

1) Among long-range mechanisms, such as global tokens and sliding-window attention, we find a simple pooling-augmented blockwise attention to be the most effective choice for various tasks.

2) For the pretraining corpus, we surprisingly find that using randomly concatenated documents from a large open-domain corpus (CommonCrawl) performs better than using existing long-document corpora such as book collections.

3) We experiment with various pretraining objectives including standard masked-span prediction (Raffel et al., 2020), primary sentence prediction (Zhang et al., 2020), and a novel model-based span prediction objective. While we find all of these objectives can bring gains over models that are not pretrained on long texts, we consider the masked-span prediction objective (using both short and long spans) remains as the best choice, thanks to its simplicity and balanced effectiveness on both short- and long-output tasks.

Using these findings, we build a strong long-context text-to-text model that establishes new state-of-the-art on five long-text summarization tasks (with $> 10\%$ relative ROUGE-2 improvements on three of the datasets) and achieves competitive performance on long-text QA tasks despite its modest size.

## 2 Model and Data

### 2.1 Efficient Models for Long Sequences

Our model is based on a standard transformer with block-sparse self-attentions (Zaheer et al., 2020) on the encoder side. While various new architectures (Wang et al., 2020; Choromanski et al., 2021; Lei, 2021; Gu et al., 2021) have been proposed, we stick to the simple architecture for the following reasons: 1) it makes it easy to reuse existing pretraining pipelines, which are often highly optimized specifically for vanilla transformers, e.g., learning rate schedules, normalization layers, optimizers; 2) using local attentions, where each token attends to only tokens in the local context, allows our model to reuse all the model parameters from existing PLMs, while other attention variants use different parameterizations that prohibit inheriting the weights of an existing pretrained model.

In addition to block attention, we investigate three mechanisms that enable long-range connec-

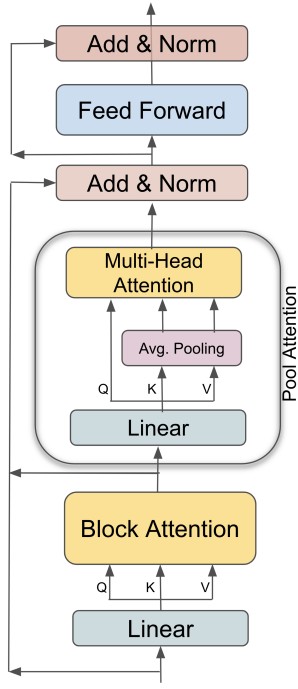

Figure 1: The pooling augmented self-attention layer. The pooling attention parameters marked separately are newly introduced and randomly initialized.

tions in the encoder:

1) ***Global-token mechanism***: Previous work (Guo et al., 2022; Zaheer et al., 2020; Beltagy et al., 2020) has proposed augmenting block-sparse attention with a small set of "global tokens" that attend to the entire sequence and hence enable long-range interactions in the encoder. Specifically, we mark the first 64 tokens in each attention block as global tokens and share the projection matrices for both the global and regular tokens. This mechanism has proven effective in encoder-only models, especially for question answering tasks as shown by the aforementioned methods.

2) ***Overlapping (strided) attention windows***: Sliding-attention with overlap is a straightforward way to introduce long-range connections in local attention models. As we stack the layers in the encoder, the receptive field of each token would increase exponentially. For example, (Beltagy et al., 2020) use the stride of one token and each token attends to an equal number of tokens from both sides. We develop a simpler and faster block-wise version which makes the parallelization easier to implement; namely, tokens in each block will attend to all the tokens inside the block, and half of the tokens from its immediate left and right blocks.

3) ***Pooling layers***: Recent work (Zhang et al., 2021; Pang et al., 2022a) has explored using pool-

ing operations to reduce the number of *key* and *value* states in transformers. We implement a simpler version that only requires standard average pooling operations. All illustration of the pooling-augmented attention layer is shown in Figure 1. Specifically, in the top n layers of the transformer encoder, we add a second attention module which takes as input the hidden states output by the $i$th block self-attention layer $\mathbf{X_i} \in \mathbb{R}^{L \times h}$, where $L$ is the sequence length and $h$ is the size of the hidden states. As in the vanilla attention layers, $\mathbf{X_i}$ is first projected to create the *key, query, value* matrices $\mathbf{Q}_i^p, \mathbf{K}_i^p, \mathbf{V}_i^p \in \mathbb{R}^{L \times h}$.[2] We first average pool the $\mathbf{K}_i^p$ and $\mathbf{V}_i^p$ sequences, with a fixed kernel/stride size, into smaller lengths $\tilde{\mathbf{V}}_i^p, \tilde{\mathbf{K}}_i^p \in \mathbb{R}^{\tilde{L} \times h}$, where $\tilde{L} \ll L$. We then apply standard attention using $\mathbf{Q}_i^p, \tilde{\mathbf{K}}_i^p$ and $\tilde{\mathbf{V}}_i^p$ resulting in $O(L \times \tilde{L})$ complexity. The output of the pooling layers is added with $\mathbf{X_i}$ to form a residual connection.

We compare these variants via the performance on downstream long-sequence tasks in Sec 3.2.

## 2.2 Pretraining Corpus

The choice of the corpus has a significant impact on the downstream results. We consider long documents from formal text domains, including Books3 (Gao et al., 2021), STORIES (Trinh and Le, 2018), RealNews (Zellers et al., 2019); and long dialogues including MediaSum (Zhu et al., 2021) and OpenSubtitles (Tiedemann, 2016). While collecting a long-document corpus seems to be a natural choice for long-sequence downstream tasks, as they are more likely to include long-range dependencies than common short texts on the internet, pretraining only on these datasets also brings the risk of overfitting to specific domains, instead of achieving consistent gains on a range of tasks. Thus, we also consider a general-domain corpus – C4 as used by T5 (Raffel et al., 2020). Additionally, instead of using randomly concatenated sequences, we also tried to concatenate semantically similar C4 documents (using similarity metric learned by dense retrieval models) with the hope that the model can learn to capture more long-range dependencies across relevant documents. We discuss the effects of these corpus variants in Sec 3.3.

---

[2]The projection layers to create these matrices are not used in existing pretrained models and will be randomly initialized before further pretraining

## 2.3 Pretraining Objectives

A variety of self-supervised pre-training objectives have been proposed for sequence-to-sequence models (Lewis et al., 2020; Raffel et al., 2020; Guo et al., 2022). In the long document setting, we ideally seek an objective that promotes long-range reasoning ability in the model. We investigate the following different pretraining objectives and the effect of input length during pretraining.

1) **T5 Span Denoising**: Applying BART's denoising objective to long sequences is computationally expensive as it requires reconstructing the entire input and incurs significant computation overhead on the decoder-side attention. Moreover, reconstructing the entire input would be at odds with most downstream tasks such as question-answering and summarization, which require generating shorter text. Thus, we adopt T5-style denoising for pretraining our model, i.e., we randomly pick a set of spans in the input sequence as the decoding target and mark them with special sentinel tokens. The model is then trained to generate the uncorrupted spans. This objective is readily applicable to long documents as we can control both the length and the number of spans. We experiment with both fixed span lengths as in (Raffel et al., 2020), and also mixed span lengths with both short and long spans, with which we hope the model is able to perform well on a range of tasks requiring differing output lengths.

2) **Pegasus – Primary Sentence Prediction**: Originally proposed for summarization pretraining in (Zhang et al., 2020) and recently used for long documents by Guo et al. (2022), this objective identifies and masks out a set of principle sentences, i.e., sentences with a high ROUGE score with the rest of the document. The model is then trained to generate these principle sentences. The output length can be controlled by choosing the number of principle sentences to mask.

3) **Model-based Denoising**: Apart from randomly selecting the decoding targets, we also explore a novel model-based objective. Here we use a separate encoder-only model (with local attention) to select decoding targets for the sequence-to-sequence model. This approach is inspired by ELECTRA (Clark et al., 2020) and we hope the prediction loss of the encoder-only model can be a good proxy to select spans that require long-range dependencies to predict. Specifically, we first mask a larger number of tokens (5,120 tokens

instead of 1,024) in the input sequence. We then apply an encoder-only masked language model to recover the masked spans. Based on the losses of the masked language model, we only keep the top 20% hard spans to train the text-to-text model. The encoder-only model can either be frozen or jointly trained with the sequence-to-sequence model.

## 3 Experiments

### 3.1 Downstream Tasks & Finetuning Setup

We evaluate the models on six summarization datasets and four QA datasets. The summarization datasets are from formal domains, including **GovReport** (Huang et al., 2021), **ArXiv & PubMed** (Cohan et al., 2018) and **BookSum Chapters** (Kryściński et al., 2021); or informal conversational domains , such as TVMegaSite & ForeverDreaming (Chen et al., 2022). For **QA**, we consider **Qasper** (Dasigi et al., 2021), which contains questions over NLP papers; **QMSum**[3] (Zhong et al., 2021), longform QA over meeting scripts, and two QA datasets on books: **QuALITY** (Pang et al., 2022b) and **NarrativeQA** (Kočiský et al., 2018).

We finetune the model with a maximum of 16,384 tokens. For long-sequence QA tasks, we adopt the input format as used by the state-of-the-art open-domain QA system (Izacard and Grave, 2021). Specifically, we repeat the question/query at the start of each attention block. We also utilize the robust finetuning technique proposed by Aghajanyan et al. (2021). We conduct a grid search over finetuning hyperparameters, such as learning rate and dropout rate, the details of which are presented in Table 8 in Appendix A. We report ROUGE[4] scores for summarization datasets. For QA, we report Exact Match (EM) scores for datasets with short answers and F1 scores for datasets with long answers.

### 3.2 Effect of Architectures

To study the effectiveness of different model choices with modest computation cost, we initialize a base-size block-attention model using BART's weights. We augment the model with three additional long-range mechanisms, as described in Sec 2.1. Note that only the pooling layers introduce additional parameters that will be randomly initial-

---

[3]QMSum is proposed as a "query-based summarization" dataset. We consider it as a special case of QA as our model uses the same input format for QMSum and other QA datasets.

[4]https://github.com/pltrdy/files2rouge

ized. Table 1 shows the results on both QA and summarization tasks. For the *global-token* mechanism, we mark the first 64 tokens of each block as global tokens. We see that pooling layers produce the most consistent improvements even for GovReport, where the baseline already achieves strong numbers. Consistent with a prior study on encoder-only models (Xiong et al., 2022), attention window overlaps fail to produce further improvements over the disjoint block-attention layers. Adding global tokens consistently helps on QA tasks but not on summarization tasks. We hypothesize that in encoder-decoder models, the cross-attention can offset the effect of global tokens, as each decoding position has access to all input tokens' representations. When finetuning our final pretrained model, we also try to combine global tokens with pooling layers for QA tasks, but we did not observe further improvements.

### 3.3 Effect of Pretraining Corpus

With the assumption that models should be exposed to as many long dependencies as possible at pretraining time, we initially tried to only pretrain the model with natural long documents that are collected from sources like books, news, and TV dialogues. However, we did not achieve consistent improvements with this corpus alone. Instead, we found it is important to include sufficient documents from diverse domains, even if those documents are mostly short sequences. We present our ablation analysis in Table 2. We reported results on small summarization datasets where the gaps are more visible. Note that the sizes of long-document corpora are usually smaller than open-domain corpus. To remove the size factor that affects model performance, we limit the pretraining steps such that the model does not see repeated examples from each corpus. The length statistics of document sources can be found in the Appendix.

We see that pretraining on corpora that only have long documents, which are often from specific domains, hurts the downstream performance for most of the datasets, except for NarrativeQA, which is from a very close domain. On the other hand, pretraining on randomly concatenated C4 documents brings visible gains for most of the tasks. In addition to directly using concatenations of random C4 documents, we tried to assemble long sequences using semantically similar C4 documents, with the hope of creating more long-range connections in

| Models | GovReport | | ArXiv | | QMSum | | Qasper | QuALITY |
|---|---|---|---|---|---|---|---|---|
| | R-1 | R-L | R-1 | R-L | R-1 | R-L | Ans F1 | Ans EM |
| block-attn baseline. | 60.5 | 57.5 | 49.0 | 44.2 | 35.2 | 30.4 | 28.0 | 31.6 |
| + attn window overlaps | 60.6 | 57.6 | 49.0 | 44.3 | 34.8 | 30.2 | 28.0 | 31.6 |
| + global tokens | 60.3 | 57.3 | 49.1 | 44.3 | 35.4 | 30.7 | 29.8 | 32.5 |
| + pooling layers | **61.0** | **58.1** | **49.1** | 44.3 | 35.9 | 31.2 | **30.6** | **32.9** |

Table 1: Ablation of different long-range mechanisms using *base-size* models.

| Models | QMSum R-1 | Qasper Ans F1 | QuALITY Ans EM | NarrativeQA Ans F1 |
|---|---|---|---|---|
| non-pretrain | 35.9 | 30.6 | **32.9** | 20.4 |
| Long corpus | 34.7 | 29.9 | 31.3 | 21.2 |
| C4 | **36.3** | **32.8** | 32.8 | **21.6** |
| C4-linked | 35.7 | 32.1 | 32.8 | 21.3 |

Table 2: Effects of pretraining corpus. Base size models pretrained for 20k steps to avoid repetitions. **Long corpus**: *Books3 + RealNews + STORIES + MediaSum + OpenSubtitles*; **C4**: *randomly concatenated documents to form long sequences.*; **C4-linked**: *concatenate related short documents using a retriever model.*

| Models | QMSum R-1 | Qasper Ans F1 |
|---|---|---|
| no-pretraining | 35.9 | 30.6 |
| + T5 avg span_len 5 - 8k | 36.7 | 32.9 |
| + T5 avg span_len 5 - 16k | 37.0 | 34.6 |
| + T5 mixed span_len | 37.0 | **35.4** |
| + pegasus | **37.4** | 34.4 |
| + model-based | 37.0 | 32.5 |

Table 3: Ablation of different pre-training objectives on C4 corpus

the pretraining sequences. For each document, we use a dense retrieval model (Izacard et al., 2021) to find similar documents and concatenate them as long pretraining sequences. We denote this corpus as "*C4-linked*". However, this new corpus is either similar or worse compared to directly using C4. We conjecture that it is because the retrieved documents may contain redundant information, making some of the masked spans trivial to predict — the training perplexity after 100k updates on "*C4-linked*" is significantly lower than that on the original C4 corpus (10.5 vs 12.2).

### 3.4 Effect of Pretraining Objectives

We compare the effects of different pretraining objectives in Table 3. The generation targets are usually paragraph-length for QMSum, while Qasper expects the model to predict spans or single sentences most of the time. All the models are pretrained for $100k$ updates on the C4 corpus. To investigate the effect of pretraining sequence length, we compare the $16k$ model with a model pretrained with $8k$ sequence length. We double the batch size

for the $8k$ length pretraining such that the input tokens in each batch stays the same. We also increase the masking ratio for the $8k$ model to $1/8$ so that the decoding sequence length remains 1,024. Note that under this setting, pretraining with $8k$-length batches is a bit slower compared to the $16k$ batches due to the decoder-side self-attention.

**Pretraining with longer sequences is useful.** While a prior work (Guo et al., 2022) pretrains their model with sequences shorter than downstream tasks, we find it is generally better to directly pretrain with longer sequences. In terms of convergence rate, we find pretraining with 8k and 16k sequences are similar (the loss curves can be found in Appendix A). For downstream results, we find that training with longer sequences lengths is indeed helpful for low-resource datasets — QMSum and Qasper are both small with a few thousand examples (*T5 avg span_len 5 - 8k* vs *T5 avg span_len 5 - 16k*). We find using a range of short spans (*mixed span_len*) tends to give more gains on QA tasks.

**Alternative objectives works similar as random masking.** While the **Pegasus** objective is effective for summarization, we do not find it to be consistently better than T5 denoising. It also incurs more data processing costs compared to T5's random masking. We also find that model-based denoising fails to yield better performance than random denoising, even though it introduces a harder pretraining task, i.e., larger training losses. We conjecture that, while this objective might provide

| Model | # Param | GovReport | | | BookSum | | | ArXiv | | | PubMed | | |
|---|---|---|---|---|---|---|---|---|---|---|---|---|---|
| | | R-1 | R-2 | R-L | R-1 | R-2 | R-L | R-1 | R-2 | R-L | R-1 | R-2 | R-L |
| BigBird | 580M | - | - | - | 31.8 | 6.5 | 14.2 | 46.6 | 19.0 | 41.8 | 46.3 | 20.7 | 42.3 |
| LED | 460M | 59.4 | 26.5 | 56.6 | 32.8 | 7.5 | 14.6 | 46.6 | 19.6 | 41.8 | 47.0 | 20.2 | 42.9 |
| PageSum | 440M | 59.9 | 27.2 | 57.1 | - | - | - | 49.7 | 21.1 | 44.7 | 48.2 | 21.1 | 44.3 |
| BART-Hepos | 440M | 56.9 | 22.6 | 53.8 | - | - | - | 48.2 | 20.3 | 41.8 | 48.1 | 21.1 | 42.7 |
| DYLE | 525M | 61.0 | 28.8 | 57.8 | - | - | - | 46.4 | 18.0 | 41.5 | - | - | - |
| LongT5-large | 750M | - | - | - | - | - | - | 48.3 | 21.6 | 44.1 | 50.0 | 24.7 | 46.5 |
| LongT5-xl | 3B | - | - | - | - | - | - | 48.4 | 21.9 | 44.3 | 50.2 | **24.8** | **46.7** |
| Top-down (AvgP) | 460M | - | - | - | 37.9 | 9.1 | 18.0$^{\dagger}$ | 48.7 | 20.7 | 43.9 | 48.3 | 21.4 | 44.2 |
| Top-down (AdaP) | 660M* | - | - | - | 38.3 | 9.2 | 18.1$^{\dagger}$ | **51.0** | 21.9 | **45.6** | **51.1** | 23.3 | 46.5 |
| BART-LS | 440M | **62.0** | **30.9** | **59.2** | **38.5** | **10.3** | **36.4** | 50.2 | **22.1** | 45.4 | 50.3 | 24.3 | 46.3 |

Table 4: Results on long-document summarization. Pipelined approaches are highlighted in gray. LED's results on GovReport are from PageSum (Liu et al., 2022). *: The AdaPool version of the Top-Down model requires an additional encoder model to predict the weights in its pooling layers. $^{\dagger}$: The baseline R-L scores on BookSum are taken from Pang et al. (2022a) and may not be rigorously comparable due to the unknown ROUGE script version used in their paper.

more training signals that are related to long-range dependencies, it can also introduce noisy supervision, which is harmful for the model to learn a wide range of language understanding skills.

## 3.5 Main Results

**Best Model Configuration.** Following the analysis of base-size models, we pretrain a large-size model with the best configuration, which consists of (a) block attention and pooling layer augmentations applied to the vanilla Transformer architecture, (b) long-sequence training data batches formed by randomly concatenating the documents from C4 corpus, and (c) T5 denoising loss with a mix of short and long spans as the training loss. We pretrain the model for 100K steps. Our model is denoted as "BART-LS" in the following sections.

### 3.5.1 Summarization

Table 4 shows results of formal long-document summarization. We first compare our model with models that directly reuse existing models' weights without further pretraining and newly introduce parameters. Apart from BigBird and LED that simply use encoder-side local attention to allow existing PLM to take longer context, we also consider more recent baselines including Page-Sum (Liu et al., 2022) which investigates the locality bias on both encoder and decoder side; BART-Hepos (Huang et al., 2021) which applies head-wise cross-attentions; DYLE (Mao et al., 2022) which combines a context extractor with a generator that only takes short text as input, and uses a complex training pipeline to provide supervision for the extractor. Our model outperforms BigBird, LED, and BARTHepos by a large margin. With simple sequence-to-sequence finetuning, our model

also consistently outperforms PageSum and DYLE which are specifically designed for summarization tasks. Note that PageSum proposes the idea of using a weighted combination of multiple decoder predictions (corresponding to taking the encodings of different parts of the input sequences as inputs), which could be orthogonal to our method.

Compared to LongT5 (large and xl), our model achieves stronger performance on ArXiv and is on-par on PubMed, even with much fewer parameters. The recently proposed Top-Down Transformer (Pang et al., 2022a) applies a similar pooling operation at the finetuning stage. Our model architecture is similar to their "Average Pooling" variant but conceptually simpler. With the proposed pretraining method, our model outperforms "Top-down (AvgP)" on all tasks. Besides "Top-down (AvgP)", the authors also proposes a more advanced pooling layer that uses the token importance predicted by another encoder-only model to aggregate the hidden states for each pooling operation, i.e., "Top-down (AdaP)". While this method should be orthogonal to our model during finetuning, we find the model-based adaptive pooling hard to replicate. Our model matches the performance of "Top-down (AdaP)" performance on ArXiv and PubMed in terms of R-2/R-L, and surpass their results on BookSum.

In contrast to formal documents, dialogue texts, especially multi-person conversations, can be noisier, more unstructured, and cover more diverse topics within each document. We test our model on two summarization datasets collected from popular TV series (Chen et al., 2022). As shown in Table 5, our model achieves even stronger relative gains compared to gains on formal-domain datasets.

| Model | TVMegaSite | | | ForeverDreaming | | |
|---|---|---|---|---|---|---|
| | R-1 | R-2 | R-L | R-1 | R-2 | R-L |
| BART-large | 43.5 | 10.3 | 41.4 | 33.8 | 7.5 | 29.1 |
| DialogLM | 45.6 | 10.8 | 43.3 | 35.8 | 8.3 | 30.8 |
| Top-down (AvgPool) | 49.3 | 14.4 | 47.5 | 35.8 | 8.9 | 31.1 |
| Top-down (AdaPool) | 51.0 | 14.7 | 49.0 | 36.8 | 9.2 | 31.1 |
| BART-LS w/o pretrain | 50.9 | 14.5 | 48.9 | 37.1 | 9.6 | 32.5 |
| BART-LS | **51.8** | **17.2** | **50.0** | **39.1** | **10.7** | **33.5** |

Table 5: Results of on long dialogue (scripts from TV series) and narrative summarization.

| Model | QMSum | | |
|---|---|---|---|
| | R-1 | R-2 | R-L |
| BART-large | 32.2 | 8.0 | 27.7 |
| DialogLM | 34.0 | 9.2 | 30.0 |
| DYLE | 34.4 | 9.7 | 30.1 |
| LED | 34.2 | 10.3 | 30.0 |
| SecEnc | 37.1 | 13.0 | 32.6 |
| SecEnc-W | 37.8 | 13.4 | 33.4 |
| Block-BART (ours) | 36.6 | 12.1 | 32.4 |
| BART-LS | **37.9** | 12.1 | 33.1 |

Table 6: Results on query-based meeting summarization (QMSum). The highlighted row indicates additional data has been used for training.

| Model | Qasper F1 | NarrativeQA EM | QuALITY EM-T/H |
|---|---|---|---|
| LongT5-base | 46.6 | 23.0 | 37.9/36.6 |
| LongT5-large | 53.3 | 27.2 | 40.6/38.6 |
| LongT5-3B | 53.1 | 29.3 | 46.0/42.1 |
| Block-BART (dev) | 38.1 | 24.1 | 35.7 |
| BART-LS (dev) | 40.6 | 25.4 | 37.6 |
| BART-LS | 48.7 | 26.2 | 37.8/34.0 |

Table 7: Test results on QA tasks. LongT5's numbers are taken from the Scrolls benchmark (Shaham et al., 2022). We also compare our model with a block-attention baseline that reuses BART's weights on the dev set, as shown in gray rows. Note that our model's size is in between LongT5-base and LongT5 large.

Note that DialogLM (Zhong et al., 2022) is specifically designed for the dialog domain and further pretrained a PLM checkpoint on dialog corpus. The large improvements over their results again suggest the importance of open-domain pretraning corpus.

### 3.5.2 QA and Query-Based Summarization

As mentioned in Sec 3.1, we use the same input format for finetuning QA tasks and query-based summarization. As there are no existing baselines of long models that reuse the weights of short-sequence models, we also report the performance of our implementation of block-attention BART. As shown in Table 6, our model outperforms all previ-ous methods that do not apply data augmentation. SecEnc (Vig et al., 2022) is also a block-attention version of BART – it distribute overlapped texts (instead of disjoint text blocks) into each self-attention window and reuses the position embeddings of the first 1,024 tokens. On long-document QA datasets (as shown in Table 7), our best model is consistently better than our block-attention baseline and is aligned with LongT5 in terms of scaling effect – our model's size is between the base and large versions of LongT5.

### 3.6 Performance Analysis on Input Lengths

To further investigate the performance gains of our proposed model, we compare the performance of the proposed model against the base model as a function of source document length for two summarization datasets, namely SummScreen and TVMegaSite. To conduct our analysis, we divide the validation split of both the datasets into *short* and *long* documents. The cutoff length to separate the two groups is chosen such that approximately 75% of the documents are classified as short documents. Figure 2 presents the results of this comparison. For both the datasets: (a) there's a performance drop for both the best and the base model for longer documents, and (b) the best model is better than the base model on all data splits. For SummScreen the performance gap between the best and the base model is bigger for *long* documents than for *short* documents – relative ROUGE-L increase of 0.80% and 3.96% for *short* and *long* documents respectively. This suggests that the performance gains for the best model can be attributed to better long-context modeling. For TVMegaSite this trend of increasing performance gap between the best and the base model with an increase in document length still holds true, though the increase in performance gap is modest in comparison to the increase observed for SummScreen – relative ROUGE-L increase of 2.43% and 2.75% for *short* and *long* documents respectively.

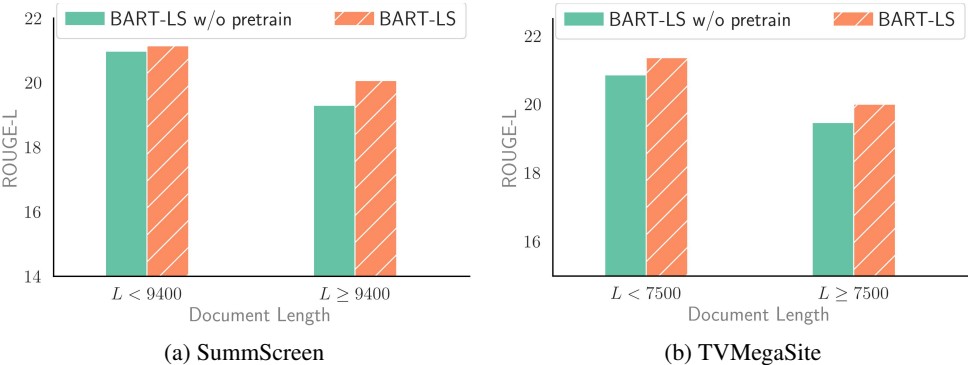

|                    |                    |
|:------------------:|:------------------:|
| (a) SummScreen     | (b) TVMegaSite     |

Figure 2: ROUGE-L scores as a function of source document length for the base model and the best model for two dialogue summarization datasets.

## 4 Related Work

### 4.1 Efficient Transformer Architectures

A long list of works has been proposed to reduce the complexity of the attention layers of transformers. The simplest paradigm is to restrict each token's attending context to a subset of the whole sequences, e.g., Reformer (Kitaev et al., 2020) and the Routing transformer (Roy et al., 2021) proposes hashing or clustering based attention, where each token only attends to tokens of a single bucket/cluster. Our model architecture is influenced by previous work like Longformer (Beltagy et al., 2020), BigBird (Zaheer et al., 2020) and ETC (Ainslie et al., 2020) that demonstrate strong downstream performance. These models assume strong locality bias in language data and restrict each token's attending context to nearby tokens. In contrast, we augment the block attention with pooling layers and study the effect of additional pretraining on long sequences. Other approaches tackling the efficiency bottleneck includes kernel-based (Choromanski et al., 2021; Peng et al., 2021) and low-rank approximation (Wang et al., 2020) of the N×N attention matrix. However, in contrast to local attention transformers, the effectiveness of these approximation approaches is yet to be validated in downstream tasks.

### 4.2 Generation from Long Text Inputs

To apply pretrained models to long-sequence tasks, early studies (Zaheer et al., 2020; Beltagy et al., 2020) reuse parameters from models pretrained on short sequences and replaces the encoder full attention with sparse local attentions. While the models are not exposed to long sequences at pretraining time, they demonstrates consistent improvements over previous models that can only take truncated inputs. Complementary to local attentions, Zhang et al. (2021) show that pooling layers can

be inserted into a pretrained transformer at finetuning time and bring additional performance gains on summarization. Instead of relying on a single model that directly processes the whole input, Mao et al. (2022) proposes a two-stage extract-and-generate approach, where the extractor can leverage the supervision signal learned by the generator. However, despite the complicated training recipe, it does not bring consistent gains and underperforms our non-pretrain baselines. The most relevant work to ours is LongT5 (Guo et al., 2022), which adopts both global tokens as well as local attention, and pretrains the model with 4k text sequences from C4. Compared to LongT5, we augment local attentions with pooling layers and present a more comprehensive study on pretraining strategies. Without pretraining from scratch, we achieve stronger summarization performance. Concurrent to our work, Phang et al. (2022) also present an empirical study on adapting short-text models for long document summarization. While their study mostly focuses on architectures, we present additional analysis on the choices of pretraining corpus and learning objectives.

## 5 Conclusion

Through a comprehensive study on the effects of model architectures, training losses and pretraining dataset, we present an effective recipe to adapt existing pretrained text-to-text models for long-sequence NLP tasks. The resulting model sets new state-of-the-art on five long-sequence summarization tasks and achieves consistent gains on QA over local-attention models that simply reuse BART's parameters. Apart from presenting a stronger checkpoint for finetuning on downstream tasks, we hope our findings in the study can provide insights for future works that aim to develop stronger long-sequence models for downstream tasks.

## 6 Limitations

Pretraining language models is a costly endeavor, and even more so in the case of long-context PLMs. Because of computational budget constraints, we only explored a limited space of the hyperparameter search space.

- We experiment with training on either just long document corpora or a pseudo long document corpora formed by concatenating short documents. Future work can investigate using a combination of the two.

- We have a surprising empirical finding that pretraining on *pseudo* long documents formed by concatenating random documents of a short-document corpora (C4) outperforms both: (a) pretraining on actual long documents from a long-document corpora, and (b) pretraining on *pseudo* long documents formed by concatenating related documents from the same short-document corpora. Future work can investigate in more detail the reasons for these empirical gains, and also test these models on their discourse understanding.

- Due to the human evaluation cost for long-context summarization tasks, we rely on automatic metrics which can be unreliable as suggested by prior work (Kryscinski et al., 2019; Fabbri et al., 2021).

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

## A  Additional experiment/data info

**Build the linked C4 corpus**   We attempt to use text retrieval techniques to assemble long text sequences with the hope that the model can learn more long-range dependencies from linked relevant documents. We first encode all the documents into dense vectors with the Contriver (Izacard et al., 2021) encoder. For documents that have more than 512 tokens, we use primary sentences (Zhang et al., 2020) as the input to the encoder. Directly retrieving documents from the whole index (340M vectors) is prohibitive in terms of computation cost. We follow the idea inverted indices, we first k-means to get 256 clusters of documents and then assemble long sequences within each cluster. Starting from each documents, we concatenate it with its top-k nearest neighbors until the length exceeds certain threshold. To avoid repeated documents,

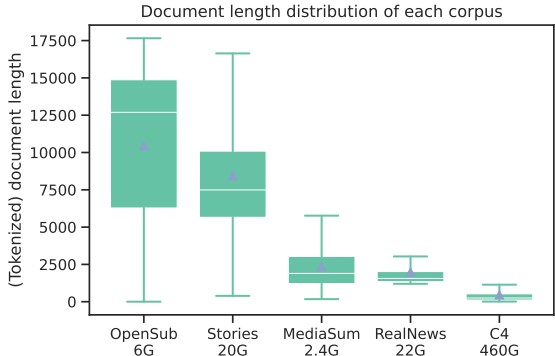

Figure 3: Document length distribution of each source corpus. The sizes of each corpus (file sizes of tokenized texts) are also shown in the x-axis. The median and mean lengths are denoted via the while line and the triangle. We did not show the statistics of the Books3 corpus (60G) here as it has much longer documents with mean/medium over 100k tokens.

we enforce that each documents can appear in at most 2 sequences.

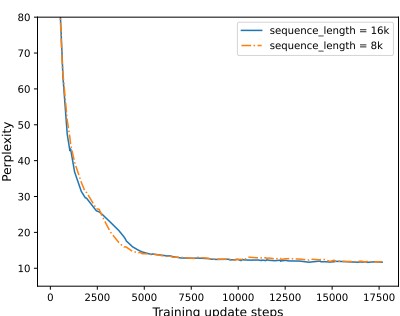

Figure 4: Training curves with 8k/16k sequence lengths. Pretraining with different sequence lengths shows similar level of data efficiency.

**Hyperparameters** We use a fixed set of hyperparameters for pretraining: we set the learning rate to be $1e-4$, the weight decay coefficient to be $0.01$ and applies polynomial decay with $500$ warm up steps; we use a batch size of $256$ (16,384 tokens per sample) and fix the random seed to $42$. The hyperparameter grids for the downstream tasks are shown in Table 8.

| Downstream Task | learning rate | batch size | max epoch | dropout | warmup steps (polynomial lr decay) |
|---|---|---|---|---|---|
| arXiv | 1e-4, 3e-4, 4e-4 | 128 | 8 | 0, 0.1 | 200 |
| GovReport | 5e-5, 3e-4, 4e-4 | 128 | 70 | 0, 0.1 | 200 |
| PubMed, BookSum | 3e-4, 4e-4 | 64 | 60 | 0, 0.1 | 200 |
| SummScreen | 5e-5, 3e-5, 1e-4 | 64 | 130 | 0, 0.1 | 200, 500, 1000 |
| Qasper, QMSum, Quality | 1e-4, 5e-5, 3e-5 | 32, 64 | 150 | 0, 0.1 | 100, 200 |
| NarrativeQA | 5e-5, 3e-5 | 64 | 8 | 0, 0.1 | 200 |

Table 8: Hyperparamter grid for downstream task finetuning. We use Adam optimizer ($\beta$ = (0.9, 0.999), $\epsilon$ = 1e-6) for all tasks.

| Downstream Task | generation parameters |
|---|---|
| arXiv | beam: 4, max_len: 300, min_len: 50, length_penalty: 5.0, no_repeat_ngram: 3 |
| GovReport | beam: 4, max_len: 740, min_len: 50, length_penalty: 4.0, no_repeat_ngram: 3 |
| PubMed | beam: 4, max_len: 400, min_len: 40, length_penalty: 4.0, no_repeat_ngram: 3 |
| BookSum | beam: 4, max_len: 550, min_len: 20, length_penalty: 4.0, no_repeat_ngram: 3 |
| SummScreen-FD | beam: 4, max_len: 300, min_len: 50, length_penalty: 4.0, no_repeat_ngram: 3 |
| SummScreen-TVM | beam: 4, max_len: 640, min_len: 50, length_penalty: 5.0, no_repeat_ngram: 3 |
| Qasper | beam: 4, max_len: 80, length_penalty: 1.0, no_repeat_ngram: 3 |
| NarrativeQA | beam: 4, max_len: 20, length_penalty: 3.0, no_repeat_ngram: 3 |
| QMSum | beam: 4, max_len: 256, min_len: 40, length_penalty: 4.0, no_repeat_ngram: 3 |
| QuALITY | beam: 4, max_len: 50, length_penalty: 3.0, no_repeat_ngram: 3 |

Table 9: Generation parameters for each task.