# OpenReview forum: "Adapting Pretrained Text-to-Text Models for Long Text Sequences"
_EMNLP/2023/Conference — EMNLP 2023 Findings_

### Official Review · Reviewer_kAsi · 2023-08-01

**Soundness:** 3

**Excitement:**

3: Ambivalent: It has merits (e.g., it reports state-of-the-art results, the idea is nice), but there are key weaknesses (e.g., it describes incremental work), and it can significantly benefit from another round of revision. However, I won't object to accepting it if my co-reviewers champion it.

**Paper Topic And Main Contributions:**

This paper presents a method for adapting text-to-text models to handle longer text sequences. The authors conducted an empirical study of three aspects of the pretraining pipeline: the model architecture, the optimization objective, and the pretraining corpus. They proposed an approach to building long-context models from existing short-context models. The resulting model achieved competitive results on summarization tasks.

**Reasons To Accept:**

A valuable empirical study of methods for long-text modeling

**Reasons To Reject:**

Although this paper explores some methods, I have a hard time understanding its most significant contributions. It explores many aspects (model, data, objective) but fails to give an in-depth analysis of each aspect. For example, its modeling exploration is more like an empirical study of existing methods. I noticed that this paper claims it has some interesting findings, but I don't think it shows sufficient insight that can help the community better understand or solve the long-text modeling challenge.

Also, the mainstream research focusing on long-text modeling is mostly about the GPT architecture now. While it makes no sense to follow the mainstream research to also study this problem on the GPT-style models, this paper should add more discussion about the recent related work because I think there are a lot of methods that can be shared regardless of enc-dec models or decoder-only models.

**Reproducibility:**

3: Could reproduce the results with some difficulty. The settings of parameters are underspecified or subjectively determined; the training/evaluation data are not widely available.

**Reviewer Confidence:**

3: Pretty sure, but there's a chance I missed something. Although I have a good feel for this area in general, I did not carefully check the paper's details, e.g., the math, experimental design, or novelty.

---

### Official Review · Reviewer_eLL5 · 2023-08-04

**Typos Grammar Style And Presentation Improvements:** N/A
**Soundness:** 3

**Excitement:**

4: Strong: This paper deepens the understanding of some phenomenon or lowers the barriers to an existing research direction.

**Missing References:**

N/A

**Paper Topic And Main Contributions:**

This paper presents a study on adapting pretrained text-to-text models for long text sequences. The authors propose a recipe for building long-context models by replacing full attention with pooling-augmented blockwise attention and pretraining the model with a masked-span prediction task. They also compare the effectiveness of using randomly concatenated short documents versus existing long document corpora for pretraining. The findings show that the proposed approach achieves competitive performance on long-text QA tasks and sets a new state-of-the-art on long-text summarization datasets.

**Questions For The Authors:**

Please address the concerns mentioned in "Reasons to Reject".

**Reasons To Accept:**

1. The paper presents a comprehensive study on adapting existing pretrained text-to-text models for long-sequence NLP tasks, offering valuable insights and findings for the research community. The authors explore the effects of model architectures, training losses, and pretraining datasets, providing an effective recipe for building high-performing long-context models.

2. The proposed model achieves state-of-the-art performance on five long-sequence summarization tasks and consistently outperforms local-attention models that simply reuse BART's parameters. This demonstrates the effectiveness of the adapted model and its potential for improving downstream tasks such as question answering.

3. The paper conducts performance analysis on input lengths, comparing the proposed model against the base model for short and long documents. The results show that the proposed model performs better than the base model on all data splits, with a larger performance gap for long documents. This suggests that the model's performance gains can be attributed to better long-context modeling, providing further evidence of its effectiveness.

**Reasons To Reject:**

1. Lack of novelty or contribution: The paper may not present significant new findings, insights, or approaches that advance the field of natural language processing. If the paper's contribution is deemed incremental or does not sufficiently differentiate itself from existing work, it may be rejected.

2. Methodological flaws or limitations: The paper may have methodological issues, such as flawed experimental design, insufficient evaluation metrics, or inadequate comparison with existing approaches. If the methodology is not rigorous or the results are not convincing, it may lead to rejection.

3. Poor presentation or writing quality: The paper may have significant issues with clarity, organization, or writing quality. If the paper is difficult to understand, lacks coherence, or contains numerous grammatical or typographical errors, it may be rejected.

**Reproducibility:**

1: Could not reproduce the results here no matter how hard they tried.

**Reviewer Confidence:**

3: Pretty sure, but there's a chance I missed something. Although I have a good feel for this area in general, I did not carefully check the paper's details, e.g., the math, experimental design, or novelty.

---

### Official Review · Reviewer_Z5kL · 2023-08-07

**Soundness:** 3

**Excitement:**

4: Strong: This paper deepens the understanding of some phenomenon or lowers the barriers to an existing research direction.

**Paper Topic And Main Contributions:**

* The paper presents a comprehensive study on how to adapt existing pretrained language models for long sequence tasks. It considers techniques that fall under three different axes - model architecture, modeling objectives and pretraining corpus.
* For model architecture - augmenting block wise attention with pooling layers results in best performance.
* For the modeling objectives, T5 style demonising objective with a mix of short and long spans achieves the best result and for the pretraining corpus, the paper reports that using randomly concatenations passages from a large open-domain corpus gives better performance than using long document corpora like books.
* Using these findings, a strong long context language model is trained that achieves strong results on long-text summarization and QA tasks.

**Questions For The Authors:**

Please see the reasons to reject section.
A: Have you done any qualitative/human analysis of the generations from the models for the different pretraining corpora?
B. Is it possible to apply any of these techniques to decoder only models for long context summarization/QA? Especially in the distractor setting in multi-hop QA where we have 10 context passages that the model has to parse through, would any of these methods help decoder only models where context window is a bottleneck?
C: How do these methods compare to techniques such as RoPE and Alibi?

**Reasons To Accept:**

* Addresses a very relevant topic to the NLP community and presents a comprehensive evaluation of various strategies that can be applied to many different models.
* Clearly written and easy to follow.
* Strong empirical performance improvements over considered baselines.


**Reasons To Reject:**

* The result that randomly concatenating passages from an open-domain corpus gives better performance than natural long form text and semantically linked passages is counter-intuitive. I would like to understand if this conclusion is an artifact of the datasets for long form training or the tasks considered. I would like to see some qualitative analysis/human evaluations done on a small subset of the generations between the different models to make sure that the conclusion is valid.
* It would have been nice to consider baselines such as Rope and Alibi relative positional embeddings to verify the performance improvement obtained by making the changes suggested in the paper.

**Reproducibility:**

4: Could mostly reproduce the results, but there may be some variation because of sample variance or minor variations in their interpretation of the protocol or method.

**Reviewer Confidence:**

4: Quite sure. I tried to check the important points carefully. It's unlikely, though conceivable, that I missed something that should affect my ratings.

---

### Meta-Review · Area_Chair_uZ4S · 2023-09-13

**Recommendation:** 3

**Metareview:**

This paper presents a comprehensive study on adapting pre-trained text-to-text models for long text sequences. It explores three key aspects: model architecture, modeling objectives, and pretraining corpus. The authors recommend replacing full attention with pooling-augmented blockwise attention, using a T5-style denoising objective with short and long spans, and pretraining with randomly concatenated passages from the C4 corpus. These adjustments result in a strong long-context language model that excels in long-text summarization and QA tasks. The work requires further study on the claim that "randomly concatenating passages improves over long documents".

---

### Decision · Program_Chairs · 2023-10-07

**Decision:**

Accept-Findings

**Comment:**

This paper presents a comprehensive study on adapting pre-trained text-to-text models for long text sequences. It explores three key aspects: model architecture, modeling objectives, and pretraining corpus. The authors recommend replacing full attention with pooling-augmented blockwise attention, using a T5-style denoising objective with short and long spans, and pretraining with randomly concatenated passages from the C4 corpus. These adjustments result in a strong long-context language model that excels in long-text summarization and QA tasks. The work requires further study on the claim that "randomly concatenating passages improves over long documents".